# Effects of Vanadyl Complexes with Acetylacetonate Derivatives on Non-Tumor and Tumor Cell Lines

**DOI:** 10.3390/molecules26185534

**Published:** 2021-09-12

**Authors:** Valentina Boscaro, Alessandro Barge, Annamaria Deagostino, Elena Ghibaudi, Enzo Laurenti, Domenica Marabello, Eliano Diana, Margherita Gallicchio

**Affiliations:** 1Dipartimento di Scienza e Tecnologia del Farmaco, University of Turin, Via P. Giuria 9, 10125 Torino, Italy; valentina.boscaro@unito.it (V.B.); alessandro.barge@unito.it (A.B.); 2Dipartimento di Chimica, University of Turin, Via P. Giuria 7, 10125 Torino, Italy; annamaria.deagostino@unito.it (A.D.); elena.ghibaudi@unito.it (E.G.); enzo.laurenti@unito.it (E.L.); domenica.marabello@unito.it (D.M.); eliano.diana@unito.it (E.D.)

**Keywords:** vanadyl complexes, tumor cell lines, intracellular localization, cell cycle, MAPKs

## Abstract

Vanadium has a good therapeutic potential, as several biological effects, but few side effects, have been demonstrated. Evidence suggests that vanadium compounds could represent a new class of non-platinum, metal antitumor agents. In the present study, we aimed to characterize the antiproliferative activities of fluorescent vanadyl complexes with acetylacetonate derivates bearing asymmetric substitutions on the β-dicarbonyl moiety on different cell lines. The effects of fluorescent vanadyl complexes on proliferation and cell cycle modulation in different cell lines were detected by ATP content using the CellTiter-Glo Luminescent Assay and flow cytometry, respectively. Western blotting was performed to assess the modulation of mitogen-activated protein kinases (MAPKs) and relevant proteins. Confocal microscopy revealed that complexes were mainly localized in the cytoplasm, with a diffuse distribution, as in podocyte or a more aggregate conformation, as in the other cell lines. The effects of complexes on cell cycle were studied by cytofluorimetry and Western blot analysis, suggesting that the inhibition of proliferation could be correlated with a block in the G2/M phase of cell cycle and an increase in cdc2 phosphorylation. Complexes modulated mitogen-activated protein kinases (MAPKs) activation in a cell-dependent manner, but MAPK modulation can only partly explain the antiproliferative activity of these complexes. All together our results demonstrate that antiproliferative effects mediated by these compounds are cell type-dependent and involve the cdc2 and MAPKs pathway.

## 1. Introduction

Vanadium is the 21st most abundant element in the outer regions of our planet, as it is present in the Earth’s crust, the water reservoirs, and the atmosphere [1]. It is essential for several species, but its role as micronutrient in humans has yet to be established [2]. The antineoplastic effects of vanadium salts were first shown in 1965 [3], even though they were found to be inactive when in 1967 they were tested among other metals to evaluate their effect against spontaneous mice tumors. The discovery of cisplatin anticancer activity increased the interest towards metals and in particular vanadium: English et al. demonstrated that vanadium inhibited terminal differentiation of murine erythroleukemia cells [4]. Moreover, Thompson et al. found that the dietary administration of vanadium inhibited chemically induced mammary carcinogenesis [5]. Vanadium also showed other biological effects: it had insulin-like activity [6] and reduced hyperlipidemia and hypertension. Since it has few adverse effects [7], vanadium has a good therapeutic potential. Several antitumor mechanisms have been hypothesized. Vanadium had a biphasic effect on cell proliferation: Faure et al. demonstrated that peroxovanadium complexes reduced the proliferation rate of neuroblastoma NB41 and glioma C6 cell lines [8], whereas Krady et al. showed that peroxovanadates formed in situ led to C3H10T1 mouse fibroblasts proliferation [9]. This biphasic, concentration-dependent effect has also been reported for in vitro tumor colony growth: vanadium salts at low (<10^−10^ M) concentrations stimulated, and at higher (>10^−10^ M) concentrations inhibited, colony formation in human tumors [10]. The anti-proliferative effects of vanadium compounds on normal and malignant cell lines were exerted mainly through cell cycle arrest, blocking the G2–M transition of the cell cycle in cancer cells [8]. Cell cycle arrest by vanadium complexes could be mediated by inhibition of protein tyrosine phosphatases, which in turn dephosphorylate subunits of the cyclin-B complex [11], and by activation of the mitogen-activated protein kinases (MAPKs superfamily) signaling pathway. V(IV) activates p38 MAPK and induces the transcription of nuclear transcription factor-kB (NF-kB), a factor involved in both cell cycle progression and apoptosis [12,13,14,15]. Vanadium could induce apoptosis through the production of reactive oxygen species in cytosol and mitochondria, which results in mitochondrial damage and cytochrome C release and subsequent activation of caspases [16,17]. 

Vanadium could also influence the invasive and metastatic potential of tumor cells, regulating cell–substrate adhesion, cell–cell contact, and actin cytoskeletal changes [6]. 

In 2013, Sgarbossa et al. synthetized and fully characterized six vanadyl complexes with acetylacetonate derivates bearing asymmetric substitutions on the β-dicarbonyl moiety [18]. All complexes exhibited good stability in solution and displayed a square pyramidal geometry with two ligands in the equatorial plane. These complexes dissolved in DMSO, coordinating a solvent molecule in the axial position; these adducts were stable when dissolved in DMSO, while, when diluted in water, the complexes were partially destabilized, and ligand replacement/rearrangement processes occurred, although the oxidovanadium (IV) moiety was never set free in solution [19]. Preliminary test on normal and tumor cells showed that these complexes were effective in inhibiting cell viability and that this activity could not be uniquely ascribed to the oxidovanadium (IV) moiety: in normal cells, hTERT-HME1, and podocytes, the antiproliferative effect seemed mainly related to the vanadyl moiety, whereas the responsiveness of tumor cells HCT 116 and HT-29 seemed to be mediated by the ligands’ properties [18]. In particular, complexes a and b (the ligands of which were a phenyl derivate) and complexes c and d (the ligands of which were a naphthyl derivate) were the most potent in inhibiting cell proliferation (Figure 1).

The aim of this work was to investigate the antiproliferative activity of complex c and d; we chose these two complexes because the oxidovanadium (IV) was functionalized with fluorescent ligands, which allowed us to study the intracellular localization of complexes. Moreover, to investigate the antiproliferative mechanism of these complexes, we studied cell cycle modulation and MAPKs activation. To perform these studies, we utilized four different cell lines. HCT 116 and HT-29 are two colon cancer cell lines, hTERT-HME1 is a non-transformed epithelial cell line of human breast origin immortalized with hTERT, while the last one is an immortalized human podocytes cell line. We chose two tumoral and two non-tumoral cell lines with the aim of demonstrating the possibly different behavior among them. Moreover, we chose as non-tumoral two cell lines of breast and renal origin, respectively, as non-tumoral colon cancer cell lines are not available for experimental use. We studied the antiproliferative activity of these vanadyl compounds in colorectal adenocarcinoma because intestinal cells are sensitive to metal compounds. One of the most effective drugs utilized in colorectal cancer therapy is oxaliplatin, a diaminocyclohexane platinum compound routinely utilized in the treatment of patient with this pathology [20,21]. Other vanadium compounds have demonstrated activity on the vitality of colon cancer cell lines, demonstrating their sensitivity to this metal [22].

## 2. Results

### 2.1. Intracellular Localization of Vanadyl Complexes

In hTERT-HME1 and HCT 116, complex d was respectively detected as a few small green and a few slightly larger granules in the cytoplasm of almost all the cells analyzed. Complex c was present in both cell lines in small quantity and in a very low percentage of cells as small aggregates. Both ligands were present in very low quantity in a few cells as very small granules, either in hTERT-HME1 or HCT 116 (Figure 2).

Both complexes spread out in podocyte cytoplasm as a diffuse reticulum in cells treated with complex c and with a prevalent plasma-membrane localization for complex d (Figure 2); ligands were not present in high quantity in these cells.

Finally, in HT-29, both complexes and ligands were present in cytoplasm but with different distributions. In fact, complex c was present as granules or aggregates primarily localized around the nuclear membrane, while complex d showed a more diffuse localization, with some granules and probably also an intranuclear localization. Ligand c spread out in the cytoplasm of all cells, while ligand d was mainly present as cytoplasmic granules (Figure 2).

### 2.2. Effect of Vanadyl Complexes on Cell Cycle 

To evaluate if vanadyl complexes were able to modify cell cycle, cells were treated with complex c and d for 24 h at the IC_50_ obtained from proliferation assay, and their activity was compared with that of VOSO_4_ and ligands c and d (Figure 3a,b). 

Cell cycle distribution was evaluated by cytofluorimetry, as described in Materials and Methods (4.5). Treatment with complex d for 24 h induced a block in S-G2/M phase of the cell cycle in hTERT-HME1 and in podocytes, but in the last cell lines there was no statistical significance. The proportion of cells in the G1 phase decreased in all the cell lines analyzed but HT-29 (Figure 3a,b). 

In hTERT-HME1, podocytes, and HCT 116, the S and G2/M populations increased relative to control, but the difference was statistically significant only for the G2/M phase in HCT 116. In all cell lines analyzed, complex c induced less variation in cell cycle phase distribution than complex d. 

To better understand the mechanisms of cell cycle variation induced by vanadyl complexes, the modulation of phosphorylation of two proteins involved in cell cycle progression, cdc2 and p53, was investigated (Figure 4a,b). 

Both complex c and d induced phosphorylation of cdc2 in all the tested cell lines. In hTERT-HME1 and podocytes, the treatment with complex c and d or VOSO_4_, but not with ligands, increased cdc2 phosphorylation in a significant manner (*p* < 0.001). Additionally, in HCT 116, both complexes induced a significant increase in cdc2 phosphorylation (*p* < 0.05), which was lower than that induced by VOSO_4_ (*p* < 0.001); the degree of phosphorylation of this protein by the ligands was not statistically significant. In contrast, in HT-29, both complexes and ligands induced a statistically significant phosphorylation of cdc2, even though the increase was greater for complexes (complex c *p* < 0.01, complex d *p* < 0.001, ligand c and d *p* < 0.05); the treatment with VOSO_4_ did not modify cdc2 phosphorylation in a statistically significant manner. None of the treatments modulated p53 phosphorylation in podocytes, HCT 116, and HT-29. The absence of p53 phosphorylation in hTERT-HME1 (Figure 4b) is not surprising, as it has been demonstrated that this cell line carries a functionally inactive p53 [23]. 

### 2.3. Effect of Vanadyl Complexes on MAPK

To investigate the possible mechanism of action of vanadyl complexes, cells were treated for 30 min with complexes, VOSO_4_, or ligands. Complexes were used at the IC_50_ calculated from the antiproliferative assay, and ligands were used at the same concentrations of the respective complex. Cells were stimulated with VOSO_4_ at the IC_50_ calculated from the previously conducted antiproliferative assays or at the maximum concentration if the 50% of proliferation inhibition was not reached [18]. The used concentrations are reported in the table presented in Figure 5.

MAPKs modulation was cell-dependent; in hTERT-HME1, complex d significantly increased JNK activation (*p* < 0.01 vs. control) and significantly decreased p42/44 activation (*p* < 0.05) (Figure 5a,b). Neither VOSO_4_ nor complex c nor ligand d modified JNK and p42/44 phosphorylation. Surprisingly, ligand c inhibited p42/44 phosphorylation. p38 was not modulated by treatment with VOSO_4_ or complexes. On the other hand, in podocytes, JNK phosphorylation was induced by complex c (*p* < 0.01) but not by complex d; the effect observed treating cells with complex c was similar if cells were treated with ligand c (*p* < 0.01). Complex d and VOSO_4_ induced p42/44 phosphorylation (*p* < 0.001 and *p* < 0.01, respectively). Both ligands inhibited p42/44 activation, but only ligand d in significant manner (*p* < 0.05). Complexes, ligands, and VOSO_4_ all increased p38 activation (Figure 5).

Both complexes and ligands activated JNK in HT-29 (*p* < 0.05) but not in HCT 116, in which we observed a significant decrease in JNK phosphorylation (*p* < 0.05). VOSO_4_ was able to turn on p38 and p42/44 in HCT1 116, but not in HT-29; in HCT 116, complex d had an opposite effect, as it reduced p38 and p42/44 phosphorylation and complex c did not modulate the activation of these proteins. In HT-29, both complexes and ligands were able to induce p38 phosphorylation, and they did not modulate p42/44 activation. VOSO_4_ did not modify p38 and p42/44 phosphorylation in HT-29 (Figure 5).

In order to understand if the modulation of MAPKs activation influenced the antiproliferative effect of vanadyl complexes, cells were pre-treated with MAPKs inhibitors for 30 min before incubation with vanadyl complexes (Figure 6). Cells proliferation was evaluated as described in Section 4.7.

The pre-treatment with AZD6244 (p42/44 inhibitor) or with SB202190 (p38 inhibitor) reduced the antiproliferative effect of complex c only in podocytes; the IC_50_ of complex c shifted from 5.49 ± 1.07 in absence of MAPKs inhibitor to 20.16 ± 1.33 and to 44.29 ± 1.54 when cells were pre-treated with AZD6244 (100 nM) and SB202190 (100 nM), respectively (*p* < 0.001) (Table 1).

These inhibitors did not modify the antiproliferative effect of complex d in podocytes. In contrast, in HCT 116, AZD6244 and SB202190 reduced the antiproliferative effect of complex d, but not of complex c; the IC_50_ of complex d moved from 4.42 ± 1.07 to 6.81 ± 1.14 (*p* < 0.05) and 7.99 ± 1.12 (*p* < 0.01) in presence of AZD6244 and SB202190 respectively. Neither AZD6244 nor SB202190 modified the antiproliferative effect of either of the complexes in hTERT-HME1 or in HT-29. The pre-treatment of cells with the JNK inhibitor, SP 600125, did not influence proliferation of any of the studied cell lines treated with both complexes (Figure 6).

## 3. Discussion

Among metals studied because of their antitumor activity, vanadium has received a substantial amount of attention, and vanadium complexes are a group of compounds with potential pharmacological effects. Several studies demonstrated that vanadium compounds reduced tumor growth both in vitro and in vivo [24]. Sgarbossa et al. previously synthetized new vanadyl complexes with acetylacetonate derivatives bearing asymmetric substitutions on the β-dicarbonyl moiety, underling that the antiproliferative effects of these complexes on both non-tumor and tumor cell lines was correlated not only with the vanadyl moiety, but also with the ligand properties [18]. In this paper, we selected the two most effective complexes with fluorescent properties, complex c and complex d, to further analyze their antiproliferative properties. First, we analyzed the capacity of these complexes to enter into the cells. Through confocal experiments, we observed that in general, acetylacetonate complexes showed a good distribution in the cytoplasm of almost all the cells, with a different pattern of distribution: diffuse, as in podocyte for both complexes, and as aggregates, as in HT-29 for complex c. Moreover, especially for HT-29, complexes seemed to also reach the nuclear membranes. 

Then we analyzed the effect of the presence of these complexes inside the cells. Complexes induced an increase in cell population in the G2/M phase both in non-tumor and tumor cell lines, which seemed to be correlated with the modulation of cdc2 phosphorylation; in fact, cdc2 phosphorylation in Tyr15, mediated by myt1, renders it inactive, thus blocking the progression into mitosis, and the dephosphorylation of cdc2 in this position represents a critical regulatory step for cell cycle progression [25]. Liu et al. (2012) demonstrated that vanadate can cause G2/M cell cycle arrest, which is evidenced by the increase in the level of phosphorylated Cdc2 at its inactive Tyr-15 site. We can speculate that the vanadium present in compounds c and d can also be responsible for the activation of phosphorylation of cdc2 in our cells [26]. Peroxovanadates block the G2/M transition of the cell cycle on cancer cells too, leading to a significant reduction in growth rate [8]. We also evaluated the correlation between cell cycle effect and p53 modulation, as p53 phosphorylation in Ser15 and Ser20 induced by DNA damage reduces the interaction between p53 and its negative regulator, the oncoprotein MDM2 leading to cell cycle arrest and apoptosis. Complex c and d did not modulate p53 phosphorylation, so our hypothesis was that cell cycle block was due to other mechanisms, in particular to the activation of the MAPKs signaling pathway. 

We observed a cell- (and complex)-dependent MAPKs modulation; in HT-29 and podocytes, the antiproliferative activity of complexes seemed to be correlated with the activation of JNK and p38. The p38 involvement in antiproliferative effects of V(IV) has been demonstrated in human bronchial epithelial cells, in which p38 could induce the transcription of NF-kB, involved in both cell cycle progression and apoptosis [12]. The increased JNK phosphorylation, observed also when treating hTERT-HME1 with complex d, could activate FOXO, thus decreasing the expression of cdc2, which could contribute to the G2/M cell cycle block. In HT-29, VOSO_4_ did not induce JNK phosphorylation, so this effect could be related to the ligand structure rather than to the vanadyl moiety. The effect was different in podocytes, in which both VOSO_4_ and ligands modulated p38 phosphorylation. 

Vanadium salt can also activate p42/44 [6]. In HCT 116 and hTERT-HME1, p42/44 phosphorylation was reduced by complex d; the HCT 116 pre-treatment with AZD6244 moved the antiproliferative dose–response curve toward the right, significantly increasing the IC_50_.

Although HCT 116 and HT-29 are both adenocarcinoma colorectal cell lines, they show different biochemical and morphological characteristics. Many studies have been performed to analyze the differences present in different colorectal cancer cell lines, evidencing that they can be used as preclinical models, as they showed a molecular heterogeneity similar to that observed in patients [27]. For example, HT-29 and HCT 116 differ when in the presence of mutations in some proteins of the EGFR pathway, such as KRAS, BRAF, and PI3K, or in the expression of proteins, such as COX-2 [27,28]. The HT-29 cell line harbors the BRAF V600E mutation, while HCT 116 harbors KRAS-G13D and PI3K H1047 mutations [27]. Vanadates are recognized activators of the MAPK and PI3K pathways [26,29], so the different effect of complexes on proliferation evidenced in the two cell lines can be reconducted to the different status of activation of the pathway of these proteins. In HCT 116, where PI3K is mutated and so over-activated, mainly complex d induces an inhibition of JNK and p38, reducing cell proliferation. In HT-29, there is an activation of the same MAPKs, probably linked to the state of over-activation of mutated BRAF. Oxidative stress could be involved in HT-29 stimulation, in which we observed an activation of JNK and p38, even though the effect seems to be mediated by ligands and not by VOSO_4_; ligands could also influence the interaction with other targets, such as COX-2. Additionally, the different vanadyl effect on cdc2 activation in HCT 116 and HT-29 could be correlated with the different status of PI3K in the two cell lines: vanadium upregulates cdc2 [29], and in HCT 116, the effect is higher than in HT-29 due to the presence of the PI3K H1047 mutation.

The mechanism of action of vanadyl compounds is not clear; we observed that the effect is not dependent upon p53, as also demonstrated by Clark et al. [30]. ROS generation could be involved, as oxovanadium complexes inside cells exist largely in equilibrium with vanadyl V(IV) and vanadate V(V) states [30]. Moreover, vanadium compounds enter cells through their interaction with specific receptors (e.g., transferrin, albumin); in particular, transferrin receptor 1 is more expressed in HT-29, which could explain why complexes seem to be more present into the cytoplasm of this cell line [31]. Nevertheless, HCT 116 seems to be more sensitive to the cytotoxic effect, which could be in part be explained by the higher sensitivity to oxidative stress of low metastatic cells such as HT-29, compared with high metastatic cells such as HCT 116 [32]. 

Altogether, these results underline the role of the two complexes analyzed in modulating cell cycle and cell proliferation in a cell-specific manner. The two complexes have a different distribution inside the cell and a different aggregation state. The presence of the methoxy group on the complex surface changes the uptake interaction/behavior because of its higher polarity than the naphthalene backbone, which changes the complex interaction with the cellular membrane and with the water inside the cell, and consequently changes the intracellular distribution. This can be responsible for the observed differences. Furthermore, since the reported redox potential for V(IV)/V(III) couple of similar complexes [33] has been evaluated between 0.90 V and −1.05 V, and the redox potential of NAD+/NADH is −0.32 V [34], we can reasonably exclude the involvement of redox equilibria between vanadyl complex and NADH in the mechanism of action. Unfortunately, our results evidenced that the two compounds are not selective for tumoral cell lines, but these complexes represent promising compounds to be further investigated with the aim of developing more effective and safer anticancer drugs; in particular attention will be focused on the ways of increasing the selectivity towards cancer cells using nanoparticles or other drug delivery systems.

## 4. Materials and Methods

### 4.1. Reagents

The synthesis of the ligands and complexes was performed according to the protocol described by Sgarbossa and co-workers [18]. Complexes and ligands were dissolved in DMSO as a 10 mM stock solution and stored at room temperature. SB202190, AZD6244, and SP600125 were purchased from Selleck Chemical (Houston, TX); they were dissolved in DMSO as a 20 mM stock solution and stored at −20 °C. Vanadyl sulfate (VOSO_4_), 4’,6-diamidino-2-phenylindole (DAPI), Mowiol^®^ 4-88, RNase, propidium iodide, EDTA, protease inhibitor cocktails, and monoclonal anti-β-actin antibody were purchased from Sigma-Aldrich (St. Louis, MO). VOSO_4_ was freshly dissolved in medium. DMEM, DMEM-F12, RPMI 1640, and fetal bovine serum (FBS) were purchased from Aurogene (Rome, IT). EGF, insulin, hydrocortisone, penicillin/streptomycin antibiotic mixture, amphotericin B, and glutamine were purchased from Sigma-Aldrich (Milan, IT).

### 4.2. Synthesis of Ligands and Complexes

Ligand c was purchased from Sigma-Aldrich. Ligand d was synthesized according to a previously reported procedure [18]. Spectroscopic data were consistent with those reported in the literature [18]. Vanadyl complexes were synthesized according to literature; all spectroscopic data were consistent with those reported [18]. A complete physico-chemical characterization of both complexes c and d was already reported [18].

### 4.3. Cell Culture

HCT 116, HT-29, and hTERT-HME1 cell lines were purchased from American Type Culture Collection (Manassas, VA); immortalized human podocytes were kindly provided by Prof. Gianluca Miglio (University of Turin, Dept. Scienza e Tecnologia del Farmaco). Human podocytes were obtained by infection of cultures of renal cells with a hybrid Adeno5/SV40 virus and characterized as previously reported [35]. HCT 116 and HT-29 were cultured in DMEM medium supplemented with 10% FBS. Podocytes were grown in RPMI 1640 supplemented with 10% FBS. hTERT-HME1 cells were cultured in DMEM-F12 medium, supplemented with 2% FBS, 20 ng/mL EGF, 10 µg/mL insulin, and 100 µg/mL hydrocortisone. All cell culture media were supplemented with 100 units/mL penicillin, 0.1 mg/mL streptomycin, 0.25 µg/mL amphotericin B, and 2 mM glutamine.

### 4.4. Confocal Microscopy

An amount of 20,000 cells suspended in 100 µL of medium was seeded into borosilicate glasses; after 24 h, cells were twice washed with phosphate-buffered saline (PBS) 1X and then treated with complex c and d and their respective ligands. Complexes were used at the IC_50_ calculated from the antiproliferative assay (Table 1), and ligands were used at the same concentrations as the respective complex. After 60 min, glasses were twice washed with PBS 1X and fixed with paraformaldehyde 4% for 30 min. DAPI counter-staining was subsequently performed. Mowiol 4-88 was used as mounting medium. Images were taken using a confocal laser scanning microscope (TCS SP5, Leica, Germany).

### 4.5. Cell Cycle

Cell cycles were analyzed by flow cytometry. The 1.5 × 10^6^ cells were grown in a 10 cm plate for 24 h and then incubated with complex c, d, and the respective ligands or VOSO_4_ for 24 h. Complexes were used at the IC_50_ calculated from the antiproliferative assay (Table 1), and ligands were used at the same concentrations as the respective complex. Cells were stimulated with VOSO_4_ at the IC_50_ calculated from the previously conducted antiproliferative assays or at the maximum concentration if the 50% of proliferation inhibition was not reached [18]. The 1 × 10^6^ cells were then washed twice with ice-PBS and treated overnight with ethanol 70%. All samples were incubated with 100 µg/mL RNase for 1 h at 37 °C and then with 100 µg/mL propidium iodide supplemented with 5 mM EDTA on ice. All fluorescence levels were detected using the flow cytometer BD AccuriTM C6 (Bd Biosciences, Erembodegem, Belgium). After doublet exclusion, extended analysis of the DNA content and calculations of the percentage of cells in each phase of the cell cycle were performed using Flowing Software 2.5.1 (released 4.11.2013).

### 4.6. Western Blot Analysis

Cells were grown in 6-well plates for 24 h and then starved and treated with complexes, ligands, or VOSO_4_ for different times. Complexes were used at the IC_50_ calculated from the antiproliferative assay (Table 1), and ligands were used at the same concentrations as the respective complex. Cells were stimulated with VOSO_4_ at the IC_50_ calculated from the previously conducted antiproliferative assays or at the maximum concentration if the 50% of proliferation inhibition was not reached [18]. Cells were washed twice with ice-cold PBS and lysed with RIPA buffer supplemented with a protease inhibitor cocktail. The protein concentrations of cell lysates were determined using the Pierce^®^ BCA protein assay (ThermoScientific, Waltham, MA) according to manufacturer’s instructions. Cell lysates were then resolved by sodium dodecyl sulphate–polyacrylamide gel electrophoresis (SDS-PAGE) and transferred to polyvinylidene difluoride membranes (Bio-Rad, Hercules, CA, USA). The primary antibodies used for immunoblotting were anti-p42/44 MAP Kinase, anti-phospho-p42/44 MAP kinase (Thr202/Tyr204), anti-phospho-p38 (Tyr 1068); anti-p38, anti-phospho-SAPK/JNK (Thr183/Tyr185) and anti-SAPK/JNK, anti-phospho-cdc2 (Tyr15); anti-phospho-p53 (Ser15); all from Cell Signaling Technology (Danvers, MA). The primary antibodies were diluted at 1:1000 in PBS containing 0.1% Tween-20 (PBS-T) and incubated overnight at 4 °C, except for anti-phospho-p42/44, which was diluted at 1:2000. The secondary antibody utilized for all the primary antibodies reported above was horseradish peroxidase-conjugated goat anti-rabbit IgG (Cell Signaling Technology) or horseradish peroxidase-conjugated horse anti-mouse IgG (Cell Signaling Technology), according to the manufacturer’s instruction. Both the secondary antibodies were diluted at 1:2000 in PBS-T and incubated for 1 h at room temperature. To confirm the homogeneity of the protein loaded, the membranes were stripped and incubated with monoclonal anti-β-actin antibody (1:5000), 30 min at room temperature, and subsequently with horseradish peroxidase-conjugated goat anti-mouse IgG (1:2000) (Cell Signaling Technology), 1 h at room temperature. Protein bands were visualized using Western Lightning Plus-ECL (Perkin Elmer, Waltham, MA) and quantified on the films by densitometry using the ImageJ software [36].

### 4.7. Cell Viability

Cells were seeded in 100 µL complete medium at appropriate density (1500–2000 cells/well) in 96-well plastic culture plates in triplicate. The following day, cells were pre-incubated for 30 min with the irreversible MAPK inhibitors SB202190 and AZD6244, respectively, for p38 and p42/44MAPKs, or with the reversible inhibitor SP600125 for JNK; all the inhibitors were used at 100 nM. After serial dilutions (0.206–50 µM), 100 µL of each compound, ligand, and VOSO_4_ in serum-free medium was added to cells with a multichannel pipette. Vehicle and medium-only containing wells were added as controls. Plates were incubated at 37 °C in 5% CO_2_ for 6 days, after which cell viability was assessed by ATP content using the CellTiter-Glo Luminescent Assay (Promega Italia Srl, Milano, Italy). All luminescence measurements (indicated as relative light units) were recorded on a Victor X4 multimode plate reader (Perkin-Elmer, Waltham, MA, USA).

### 4.8. Statistics

Experiments were performed at least three times. Graphs were constructed, data analyzed, and IC_50_ for each compound calculated using GraphPad Prism 6. Dose–response curves were analyzed using a log (inhibitor) vs. normalized response–variable slope model. Where indicated, the results are given as the mean ± S.E.M. Statistical analyses were performed by one-way ANOVA using Dunnett’s post hoc test (GraphPad). Differences of means were considered significant at a significance level of 0.05 (* *p* < 0.05; ** *p* < 0.01; *** *p* < 0.001).

## Figures and Tables

**Figure 1 molecules-26-05534-f001:**
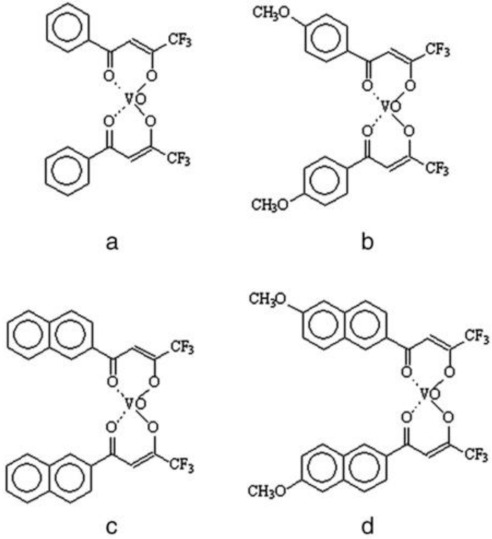
Structural formulas of complexes (**a**–**d**). (**a**) 1-phenyl-4,4,4-trifluorobutane-1,3-dione; (**b**) 1-(4-methoxyphenyl)-4,4,4-trifluorobutane-1,3-dione; (**c**) 1-(2-naphtyl)-4,4,4-trifluorobutane- 1,3-dione; (**d**) 1-(6-methoxy-2-naphtyl)-4,4,4-trifluorobutane-1,3-dione. Vanadyl complexes were synthetized and characterized as described by Sgarbossa et al. [18].

**Figure 2 molecules-26-05534-f002:**
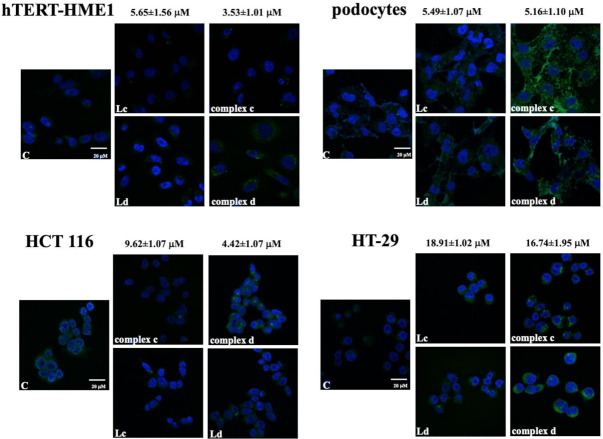
Intracellular localization of complexes c and d and their ligands (Lc and Ld). Cells were treated for 60 min with complexes used at the IC_50_ calculated from the antiproliferative assay and with ligands used at the same concentrations of the respective complex. Concentrations utilized are indicated in the figure. Images are representative of one experiment of at least four performed.

**Figure 3 molecules-26-05534-f003:**
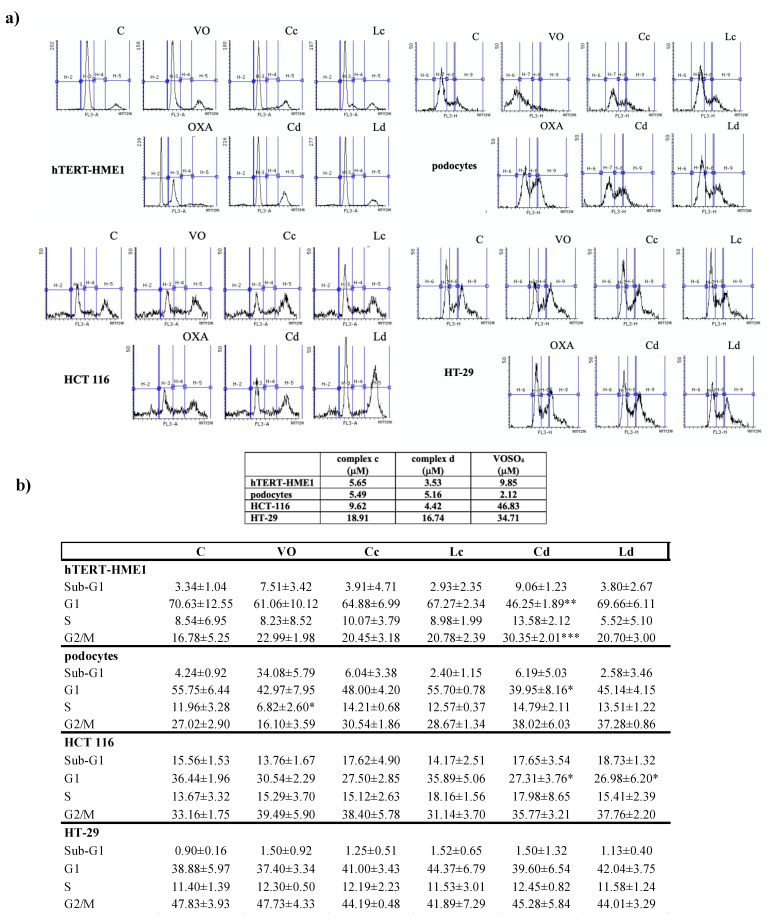
Effect of vanadyl complexes on cell cycle. Cells were treated for 24 h with complexes used at the IC_50_ calculated from the antiproliferative assay and with ligands used at the same concentrations of the respective complex. VOSO_4_ was used at the IC_50_, calculated from the previously conducted antiproliferative assays or at the maximum concentration if the 50% of proliferation inhibition was not reached. The concentrations utilized are in the table present in the figure. (**a**) The effect of complexes on cell cycle was analyzed by flow cytometry, using the flow cytometer BD AccuriTM C6 (Bd Biosciences, Erembodegem, Belgium) H-2: Sub-G1, H-3: G1, H-4: S, H-5: G2/M. (**b**) Percentage of means of at least 4 independent experiments are shown. Value are presented as the means ± S.D.; *n* = 3; * *p* < 0.05; ** *p* < 0.01; *** *p* < 0.001 vs. control (C). VO: VOSO_4_, Cc: complex c, Cd: complex d, Lc: ligand c, Ld: ligand d.

**Figure 4 molecules-26-05534-f004:**
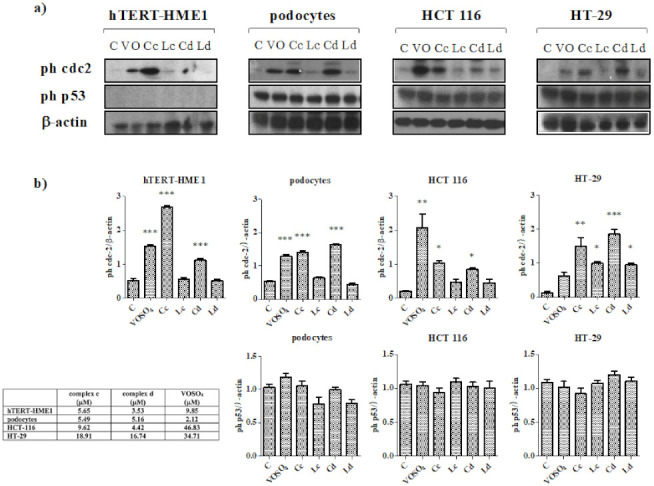
Modulation of cell cycle-related protein by vanadyl complexes. (**a**) Phosphorylation of cell cycle-related proteins was evaluated via immunoblot analysis. Cells were treated with complexes, ligands, or VOSO_4_ for 24 h. All compounds were used at the IC_50_ calculated from the antiproliferative assay, and ligands were used at the same concentrations of the respective complex. VOSO_4_ was used at the IC_50_ calculated from the previously conducted antiproliferative assays or at the maximum concentration if the 50% of proliferation inhibition was not reached. The used concentrations are reported in the table presented in the figure. β-actin was used to confirm the homogeneity of the protein loaded. Immunoblot was representative of three performed experiments. (**b**) Densitometric analyses were performed using ImageJ. Values are presented as the means ± standard deviation (S.D.); *n* = 3; * *p* < 0.05; ** *p* < 0.01; *** *p* < 0.001 vs. control (C). VO: VOSO_4_; Cc: complex c; Cd: complex d; Lc: ligand c; Ld: ligand d.

**Figure 5 molecules-26-05534-f005:**
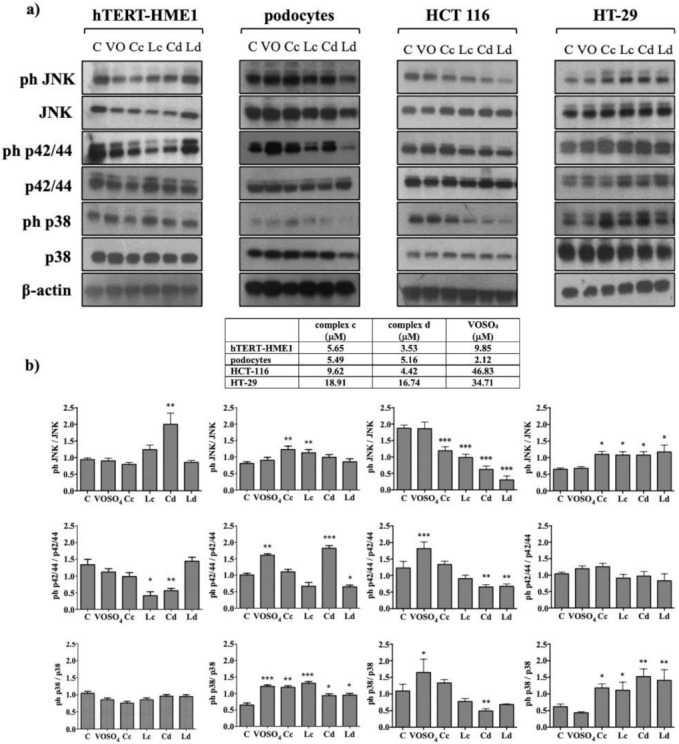
Modulation of MAPKs by vanadyl complexes. Phosphorylation of MAPKs was evaluated via immunoblot analysis. Cells were treated with complexes, ligands, or VOSO_4_ for 30 min. All compounds were used at the IC_50_ calculated from the antiproliferative assay, and ligands were used at the same concentrations of the respective complex. VOSO_4_ was used at the IC_50_ calculated from the previously conducted antiproliferative assays or at the maximum concentration if the 50% of proliferation inhibition was not reached. The used concentrations are reported in the table presented in the figure. (**a**) Phosphorylation of MAPKs proteins was evaluated via immunoblot analysis. β-actin was used to confirm the homogeneity of the protein loaded. Immunoblot was representative of three performed experiments. (**b**) Densitometric analyses were performed using ImageJ. Values are presented as the means ± S.D.; *n* = 3; * *p* < 0.05, ** *p* < 0.01, *** *p* < 0.001 vs. control (C). VO: VOSO_4_, Cc: complex c, Cd: complex d, Lc: ligand c, Ld: ligand d.

**Figure 6 molecules-26-05534-f006:**
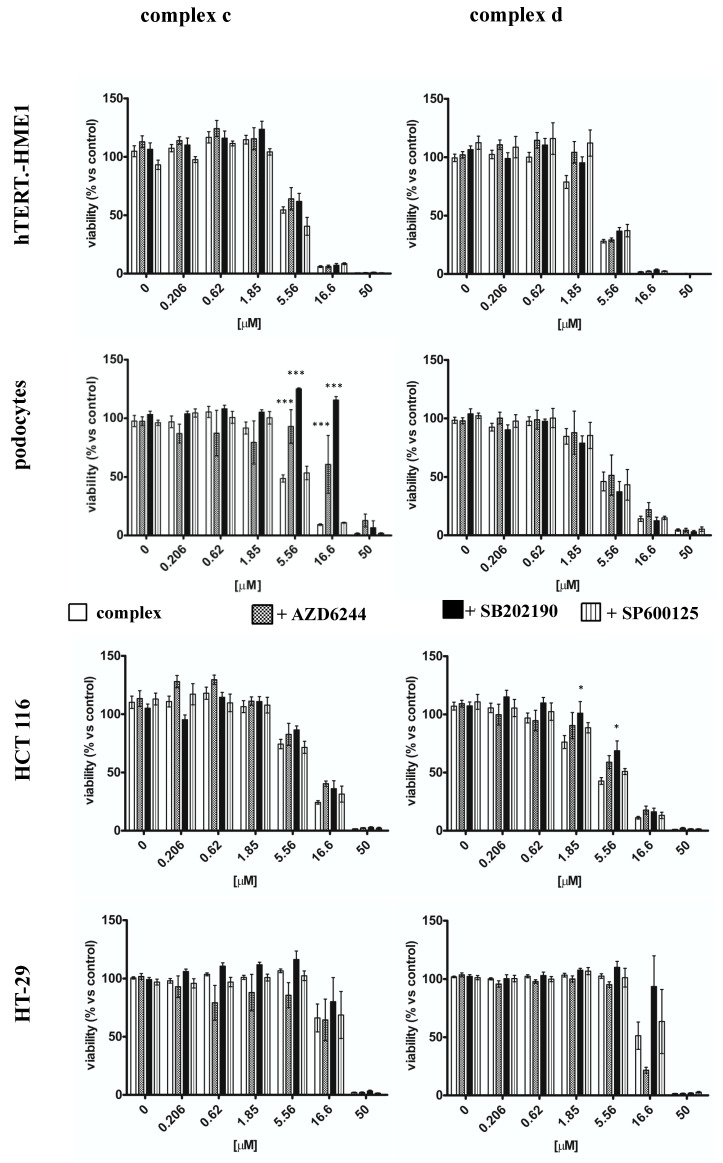
Effect of MAPKs inhibitors used in association with complex c and complex d. Cells were pre-treated with AZD6244, SB202190, or SP600125 (all 100 nM), inhibitors of p42/44, p38, and JNK, respectively, for 30 min; then cells were treated with complexes (0.206–50 μM) for 6 days. Results are normalized to the growth of non-treated cells and are represented as mean ± standard error of mean (S.E.M.) of at least three independent experiments. * *p* < 0.05, *** *p* < 0.001 vs. complex alone.

**Table 1 molecules-26-05534-t001:** IC_50_ values of antiproliferative effect of complex c and complex d. IC_50_ was calculated in all cell lines in the presence or not of MAPKs inhibitors. Cells were pre-treated with AZD6244, SB202190, or SP600125 (all 100 nM), inhibitors of p42/44, p38, and JNK, respectively, for 30 min; then cells were treated with complexes (50–0.206 μM) for 6 days. Values are expressed as mean ± S.E.M. of at least four independent experiments. * *p* < 0.05; ** *p* < 0.01; *** *p* < 0.001: complex + MAPK inhibitor vs. complex.

	**Complex c**	**+ AZD6244**	**+ SB202190**	**+ SP600125**
hTERT-HME1	5.65 ± 1.56	6.58 ± 1.12	6.58 ± 1.12	5.20 ± 1.14
Podocytes	5.49 ± 1.07	20.16 ± 1.33 ***	44.29 ± 1.54 ***	6.03 ± 1.04
HCT 116	9.62 ± 1.07	13.31 ± 1.15	13.06 ± 1.09	10.34 ± 1.13
HT-29	18.91 ± 1.02	19.87 ± 1.02	22.64 ± 1.23	19.53 ± 1.31
	**Complex d**	**+ AZD6244**	**+ SB202190**	**+ SP600125**
hTERT-HME1	3.53 ± 1.01	5.21 ± 1.89	4.71 ± 1.06	5.33 ± 1.59
Podocytes	5.16 ± 1.10	6.25 ± 1.24	4.14 ± 1.11	5.03 ± 1.06
HCT 116	4.42 ± 1.07	6.81 ± 1.14 *	7.99 ± 1.12 **	5.74 ± 1.09
HT-29	16.74 ± 1.95	11.98 ± 1.05	25.99 ± 1.45	19.20 ± 1.20

## Data Availability

Data are contained within the article.

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
