# Peer review of "Effects of Vanadyl Complexes with Acetylacetonate Derivatives on Non-Tumor and Tumor Cell Lines"

_molecules, 2021, doi:10.3390/molecules26185534_

Round 1

Reviewer 1 Report

The manuscript "Effects of Vanadyl Complexes with Acetylacetonate Derivatives on non-Tumor and Tumor Cell Lines" focuses on studying the molecular processes promoted by V(IV) compounds that lead to their antiproliferative activity. However, the authors need to provide a deeper discussion about the metal ion V(IV), ligand, and coordination compound effects on the antiproliferative and the specific enzymes and proteins activity studied.

Is the vanadyl ion participating in redox reactions?

What triggers the cdc2 phosphorylation observed when cells were exposed to vanadyl sulfate or compounds b and c? 

Both HCT116 and HT-29 are colorectal adenocarcinoma cells. How do the authors explain that the compounds c and d and their respective ligands Lc and Ld induce higher phosphorylation on HT-29 than in HCT-116?

Why does vanadyl sulfate prominently induce cdc2 phosphorylation compared with the control on HCT-116 and not in the HT-29?

Authors mention on lines 260 and 261 that Peroxovanadates block the G2/M transition of the cell cycle on cancer cells, too, leading to a significant reduction in growth rate. Thus, the authors assume that compounds c and d participate in the same redox reactions peroxovanadates.

Which is the redox potential of these compounds?

Are capable of reacting with NADH?

Which is the influence of the ligands on this redox behavior?

Much of the oxovanadium compounds upregulates or downregulates p53 phosphorylation triggering the p53/p21 pathway. Why for these compounds is not observed a change in p53 phosphorylation?

What do the authors propose to explain this phenomenon considering the multiple vanadyl compounds reported?

Does the ligand structure have a specific contribution to the mechanism of action?

If it is true, which is this contribution, with what receptor or molecules, or how do the authors hypothesize this interaction?

Quite similar compounds have shown strong interaction with DNA. The authors that conducted those studies associate their cytotoxic activities with strong DNA binding. For compounds c and d, does the DNA binding play any role in their cytotoxic activity observed?

https://www.sciencedirect.com/science/article/pii/S0141813015001026?via%3Dihub

https://pubs.rsc.org/en/content/articlepdf/2015/ra/c5ra13715b

Besides, the authors should indicate in the introduction the reason to explore the antiproliferative activity of these vanadyl compounds in colorectal adenocarcinoma. However, they studied the antiproliferative activity on the same cell lines in 2013.

Other minor comments:

Line 51: check concentration units

Line 67: use the correct symbol

Fig 1. Include compounds a and b in the figure

Line 96: Although podocytes were donated, indicate the cell line or the source

Line 119: correct section label

Line 119-121: The statement is only true for hTERT-HME1

Line 122-124: Authors mention S and G2/M population compared with control after treatment with complexes c and d, but only G2/M population of hTERT-HME1  shows the statistical difference

Line 208: correct section label

The authors need to specify the concentrations of all compounds assessed

In confocal study and western blot analysis, why use the same concentration as the used for compounds and not with the concentration that produces the maximum antiproliferative effect?

Therefore, the manuscript is not suitable for publication in its current form.

Author Response

Dear Reviewer,

you can find our responses to your comments in a file.

Reviewer 2 Report

Vanadium complexes are a group of compounds with potential pharmacological effects and antitumor activity demonstrated in vitro and in vivo. Due to their properties, these complexes are studied for the development of new generation therapeutics but their adverse effects are a major concern. This paper presents the effects of two fluorescent vanadyl complexes with acetylacetonate derivates on normal and tumoral cell lines highlighting important aspects regarding their intracellular localization, modulation of cell cycle and protein kinases activation.  However, the article requires revision and improvement of several parts before publishing.

My comments are:

Title:  the word “acetilacetonate” in the title should be replaced with “acetylacetonate”

Introduction

The author should also present several aspects of acetylacetonate derivatives to clear understand the composition of analyzed complexes. What is the role of these derivates? Are toxic effects reported in the literature?

Most of the references used by the authors are old before 2000, I suggest referring to more recent publications highlighting the latest discoveries on this subject.

Results

Figure 2. - Scale bar is missing. I suggest also, that authors enlarge the pictures with intracellular localization for better visualization of complexes and ligands. Some green dots are also visible in the control cells. Is this normal if they were unexposed to vanadyl complexes and ligands? How do the authors explain that?

I recommend that results and methods be rearranged. The cell viability test should be the first presented followed by the IC50 value determinations. Then the other evaluations can follow.

Figure 6. Why the values on X-axis are negative? I recommend changing the figures with more suggestive representations maybe bars instead of lines because many of them are superposed and error bars cant be visualized for each condition.

Materials and methods

Line 291-293: The phrase “The synthesis of the ligands….” is misplaced in the section Reagents. The authors should have a separate section about the preparation of the vanadyl complexes where to briefly describe the synthesis procedure and also to give some details about their physico-chemical characteristics. As it is known the toxicity of vanadium depends on its physico-chemical state so this is an important aspect that should be mentioned in the manuscript.

Line 323: the authors should report the concentration of paraformaldehyde

Line 326: Confocal microscopy: what kind of analyses the authors performed with the FIJI image analysis software?

Author Response

(The authors gave the same response as above.)

Round 2

Reviewer 1 Report

Most of the recommendations have been followed; however, specific points need to be resolved.  Considering that the manuscript deals with the molecular mechanism of vanadyl coordination compounds involved in their cytotoxic effects, I believe that it is essential that authors propose all the possible processes that these compounds could have within the cells (redox reactions, specific interactions with proteins, enzymes, nucleic acids, or other molecules) to explain the results.
It is clear that the authors did not perform the electrochemical studies. Still, the literature provides a broad view that could be very useful to the authors to complete this part of the discussion (response to points 2 and 5).  

doi: 10.1007/BF00926572.
http://dx.doi.org/10.1016/j.ccr.2014.12.002
10.1021/cr020607t
http://dx.doi.org/10.1016/j.canlet.2014.11.039
https://doi.org/10.1007/s12011-018-1540-6

Besides, some significant results are associated with the biochemical differences on HCT116 and HT-29 cells. Therefore, it is not enough to mention that differences exist but proposes how these differences could be involved in the different effects that vanadyl compounds produce in colorectal adenocarcinoma cells. 
The authors, in response to point 3, mentions some proteins:
HT-29 and HCT116 differ for the presence of mutations in some proteins of the EGFR pathway as KRAS, BRAF, and PI3K, or in the expression of proteins such as COX-2. 
There is no need to perform specific experiments; just conduct a full literature review to hypothesize how the vanadyl compounds could work.
The same is applicable for point 4; authors attributed different responses obtained when cells were exposed to vanadyl compounds with their different biochemical nature but did not mention which or how these differences intervein.   
In response to point 6, the authors mention: Dubouchaud et al. (2018) have calculated the NAD+/NADH redox potential in -0.32 V, then V(IV) should not be able to oxidize NADH.
https://link-springer-com.bibliopass.unito.it/content/pdf/10.1007/s10863-018-9767-7.pdf
That is true for vanadyl ions in aqueous media, but which is the effect of the ligand (theoretically because they did not perform the electrochemical analysis) in the redox potential of the compounds. The authors cannot give an exact value but could use the general information previously reported to propose a hyphothesis.
Also, they mention the inductive effect of the methoxy group will produce a slight change in the redox potential but observe a significant difference in the cytotoxic response. As the methoxy group is the only difference in the coordination compounds assessed, how do the authors believe that this substituent contributes to the cytotoxic effects if redox potential is not involved?
In response to point 9, the authors mention that the different structure of the two ligands plays a role in the mechanism of action. I agree that the authors have experimental evidence for the biodistribution and aggregation of vanadyl compounds within the cell. They have proof that the two complexes have another distribution inside the cell and a different aggregation state. I agree that the authors have experimental evidence for the biodistribution and aggregation of vanadyl compounds within the cell. However, they still do not explain or propose how the methyl group can promote or be responsible for the difference in the cytotoxic response and biodistribution difference. 
Also mention: The presence of the methoxy group on the complex surface changes the uptake interaction/behavior and, consequently, the intracellular distribution. 
Yes, that is what you observe, but How can it be done? Does this methoxy group promote the interaction with a specific membrane receptor or transporter? Are other examples reported for coordination compounds with similar behavior?
